# Guidance Gyro System with Two Gimbals and Magnetic Suspension Gyros Using Adaptive-Type Control Laws

**DOI:** 10.3390/mi16030245

**Published:** 2025-02-20

**Authors:** Romulus Lungu, Constantin-Adrian Mihai, Alexandru-Nicolae Tudosie

**Affiliations:** 1Faculty of Electrical Engineering, University of Craiova, 200585 Craiova, Romania; romulus_lungu@yahoo.com; 2International Academy of Astronautics (IAA), 75016 Paris, France; 3Doctoral School of “Politehnica” University in Bucharest, Aerospace Engineering Branch, 060042 Bucharest, Romania; adrianmihai10.07@gmail.com

**Keywords:** gyro system, missile, gimbal, magnetic suspension, DGMSGG, guidance, control

## Abstract

The authors have designed a structure for a gyro system (used for the guidance of self-guided missiles) with two gimbals and a rotor in magnetic suspension (AMBs—active magnetic bearings). The system (double-gimbal magnetic suspension gyro system for guidance—DGMSGG) orients the common axis rotor AMB (the sight line) in the direction of the target line (the guide line) by means of some control system of the gyro rotor’s rotations and translations, as well as by means of some servo systems for the gimbals’ rotation angle control. The DGMSGG provides specific signals for the missile’s autopilot, to guide it toward the target, so that the guidance line translates parallel to itself to the point of interception of the target (according to the self-guidance method by parallel approach). Based on the DGMSGG’s established mathematical model, the authors propose and design adaptive control systems for the decoupled dynamics of the gyro rotor’s translations and rotations and of the gimbals’ rotations; the concept of dynamic inversion is used, as well as linear dynamic compensators (P.D.- and P.I.D.-type), state observers, reference models, and neural networks. The theoretical results are validated through numerical simulations, using Simulink/Matlab models’ stabilization and orientation operating regimes.

## 1. Introduction

Gyro equipment and systems, in classical mechanical configuration or with magnetic suspension gyro motors (with active magnetic bearings) have a widespread use for the control of aerospace vehicles, as sensors, actuators, gyro stabilizers, or as guidance equipment. An important category is the CMG (Control Moment Gyroscope); this is an indispensable inertial actuator for the automatic attitude control systems of mini-satellites (under 500 kg), as it is suitable for fast rotational maneuvers [1,2,3,4,5,6,7]. Compared to mechanical CMGs, those with gyroscopes arranged in magnetic suspension (MSCMG), with one gimbal (SGMSCMG) or with two gimbals (DGMSCMG), possess the advantages of zero friction (thus eliminating lubrication), low noise, low vibrations, and longevity [6,8,9,10,11,12,13]. To increase efficiency, SGCMGs are used in equipment known as clusters with CMGs [1,14]. The DGMSCMGs generate two gyroscopic couples each, thus reducing the volume and the mass of the satellite. The construction of the AMB rotor allows the decoupling of its translation dynamics from the rotation dynamics and from the dynamics of the rotations of the gimbals and, implicitly, the decoupling of the control of the translation dynamics of the rotor from the control of the rotation dynamics of the rotor and of the gimbals; at the same time, it is allowed to decouple the control of the rotor translation dynamics from the control of the rotor rotation dynamics and from the control of the rotations of the gimbals [14,15,16,17,18,19,20,21,22]. To control the dynamics of translation and rotation of the AMB rotor, different linear control laws, such as P type, P.D. type, P.I.D. type, sliding mode, and back-stepping, were used [12,23,24,25,26,27].

In this paper, a guidance gyro system architecture (for self-guided missiles) with two gimbals and a rotor in magnetic suspension (DGMSGG) using adaptive-type control laws is proposed and designed.

DGMSCMG is used as an actuator in the attitude control systems of the mini-satellites, being commanded by the attitude controller through some servo systems for controlling the gimbals’ angular rates (the gyroscopic rotor reacts to the angular rates through gyroscopic moments— control moment gyros—that are transmitted to the mini-satellites and produce their rotations). Unlike the DGMSCMG, a DGMSGG (with a similar architecture to the DGMSCMG) controls the gyroscopic rotor’s direction (the line of sight) and orients it so that it overlaps with the direction of the target line (the guidance line) by means of servo systems for controlling the gimbals’ angles.

Starting from the equations that describe the nonlinear gimbals’ and gyroscopic rotor’s dynamics [16,28], the control systems for the gyroscopic rotor’s translation and rotation dynamics, as well as the control system for the gimbals’ dynamics, are designed. Compared to the papers in which the dynamics of the gyroscopic rotor translation is decoupled from the dynamics of its rotations and from the dynamics of the gyro’s gimbals [18,19,29], in this paper, all three dynamics are decoupled (the translation dynamics of the rotor from the dynamics of the rotor rotation and from the dynamics of the gimbals’ rotation) and, implicitly, one has designed three automatic control systems with superior performance to those presented in [18,20,30]. Since the rotor translation dynamics are, in fact, physically decoupled from the other two, the decoupling of the rotor rotation dynamics model from the (physically interconnected) gimbals’ rotation dynamics model leads to dynamic inversion errors, which affect the dynamic and stationary performances of the DGMSGG. That is why the control laws (based on the concept of dynamic inversion) contain, in addition to P.D. type, P.I. type and P.I.D. type components, adaptive components modeled by neural networks [29,30,31], which have the role of compensating the effects of dynamic inversion errors. The three automatic control systems include not only dynamic linear compensators, reference models, linear state observers, and neural networks but also sensors for measuring the deviation of the gyroscopic rotor, as well as transducers for measuring the gimbals’ rotation angles. The deviations of the line of sight from the guide line are determined by the target coordinator, located on the inner gimbal of the DGMSGG [31].

The paper is structured as follows. In Section 2, the DGMSGG’s architecture and functions are presented. In Section 3, the rotor’s and gimbals’ dynamics models are determined. In Section 4, the architectures of the DGMSGG’s subsystems are designed for the adaptive control of the three dynamic models. In Section 5, the results of the numerical simulations, obtained based on the Matlab R2016b/Simulink model, are presented.

## 2. DGMSGG’s Architecture and Functions

A DGMSGG has the architecture depicted in Figure 1a and consists of the following subsystems: (1) the gyro equipment with its gimbals (*e*—outer, *i*—inner) and its magnetic suspension gyroscopic rotor (*r*) [16]; (2) the target’s coordinator (CT); (3) the servo system for the control of $\sigma_{i}$ and $\sigma_{e}$ angles (the line of sight’s elevation) made up of the adaptive controller and the motor—outer gimbal and motor—inner gimbal assemblies; (4) the subsystem for the automatic control of the gyroscopic rotor’s translations $\left(x_{r},y_{r}\right)$; (5) the subsystem for the automatic control of the gyroscopic rotor’s angular deviations $\left(\alpha ,\beta \right)$; (6) the transducers for the gimbal’s rotation angles’ ($\sigma_{i}$ and $\sigma_{e}$) measurement.

The frames highlighted in Figure 1 are as follows: $Ox_{r}y_{r}z_{r}-$ related to the gyroscopic rotor; $O_{1}x_{1}y_{1}z_{1}-$, related to the stator; $Ox_{i}y_{i}z_{i}-$ related to the inner gimbal (with the $Oz_{i}-$ axis as the CT axis, the line of sight); $Ox_{e}y_{e}z_{e}-$ related to the outer gimbal. The $Oz_{T}-$ axis represents the *T*—target line (the guidance line).

The DGMSGG performs the following: (1) centering the gyroscopic rotor, which means overlapping the $Oz_{r}-$ axis (the axis of the $K_{0}-$ gyroscopic rotor’s kinetic moment) over the $Oz_{i}-$ axis of the inner gimbal (CT’s axis) by canceling both the translations (the linear deviations $x_{r}$ and $y_{r}$ of the gyroscopic rotor relative to the $Oz_{r}-$ axis) and the angular deviations α and β of the gyroscopic rotor (as Figure 1b shows), by means of the above-mentioned subsystems (having the architectures in Figure 2a,b); (2) overlaps the sight line $Oz_{i}$ over the target line, the $Oz_{T}-$ axis (canceling λ− angle’s components $\lambda_{1}$ and $\lambda_{2}$, that express the sight’s line angular deviation relative to the guidance line, as Figure 1c shows), by means of the servo system for the gimbals’ angles $\sigma_{i}$ and $\sigma_{e}$ control (having the architecture depicted in Figure 3a).

## 3. Rotor’s and Gimbals’ Dynamics Models

The models that describe the translation and rotation dynamics of the gyroscopic rotor, as well as those that describe the dynamics of the rotations of the gimbals placed on a fixed base, have been deduced and presented in [15].

The translation’s dynamics along the axis $Ox_{r}$ and $Oy_{r}$ of the magnetic suspension gyroscopic rotor (which means the use of AMB—active magnetic bearings) is described by the equations:(1a)$${\ddot{x}}_{r}=\frac{2k_{hx}}{m}x_{r}+\frac{2k_{xr}}{m}i_{x}+g_{xr},$$(1b)$${\ddot{y}}_{r}=\frac{2k_{hy}}{m}y_{r}+\frac{2k_{yr}}{m}i_{y}+g_{yr},$$
where *m* is the rotor’s mass, $k_{hx},k_{hy},k_{xr},k_{yr}-$ proportionality coefficients (for displacement—force and for current—force), $i_{x},i_{y}-$ the currents applied to the stator coils of the magnetic bearings to create electromagnetic forces along the $Ox_{r}-$ and $Oy_{r}-$ axis directions, in order to compensate for $x_{r}-$ and $y_{r}-$ linear deviations, while $g_{x_{r}}$ and $g_{y_{r}}$ are the components of the gravitational acceleration along the axes $Ox_{r}$ and $Oy_{r}$ of the rotor related frame. 

The dynamics of the rotations (angular deviations) of the gyroscopic rotor around the $x_{r}-$ and $y_{r}-$ axis is described by the following equations:(2a)$$\ddot{\alpha }=\frac{J_{rz}}{J_{rx}}\dot{\beta }{\dot{\sigma }}_{e}\sin\sigma_{i}+\frac{J_{iy}+J_{rz}-J_{iz}}{J_{rx}}{\dot{\sigma }}_{e}^{2}\sin\sigma_{i}\cos\sigma_{i}-\frac{K_{0}}{J_{rx}}\left(\dot{\beta }+{\dot{\sigma }}_{e}\cos\sigma_{i}\right)+\frac{4l_{m}^{2}k_{hy}}{J_{rx}}\alpha -\frac{M_{xi}}{J_{rx}}$$$$\ddot{\beta }=\frac{2\left(J_{iz}-J_{iy}\right)}{\lambda_{1}}{\dot{\sigma }}_{i}{\dot{\sigma }}_{e}\sin\sigma_{i}\cos^{2}\sigma_{i}+{\dot{\sigma }}_{i}{\dot{\sigma }}_{e}\sin\sigma_{i}-\frac{J_{rz}}{J_{rx}}\left(\dot{\alpha }+{\dot{\sigma }}_{i}\right){\dot{\sigma }}_{e}\sin\sigma_{i}+\frac{K_{0}}{J_{rx}}\left(\dot{\alpha }+{\dot{\sigma }}_{i}\right)+$$(2b)$$+2l_{m}^{2}k_{hx}\frac{\lambda_{1}+J_{rx}\cos^{2}\sigma_{i}}{\lambda_{1}J_{rx}}\beta +2l_{m}^{2}k_{xr}\frac{\lambda_{1}+J_{rx}\cos^{2}\sigma_{i}}{\lambda_{1}J_{rx}}i_{\beta }-\frac{M_{ye}}{\lambda_{1}}\cos\sigma_{i}$$
where $J_{rx},J_{ry},J_{rz},J_{iy},J_{iz},J_{ey}$ are the inertia moments of the rotor, of the inner gimbal, and of the outer gimbal with respect to the axes $Ox_{r},Oy_{r},Oz_{r},Oy_{i},Oz_{i},Oy_{e}$; $l_{m}$ is the distance from the origin *O* of the magnetic centers of the magnetic bearings’ coils; $\lambda_{1}=J_{ey}+J_{iy}\cos^{2}\sigma_{i}+J_{iz}\sin^{2}\sigma_{i}$; $M_{xi}$ and $M_{ye}$ are the torques created by the correction motors, $M_{xi}=k_{xi}i_{xi}$, $M_{ye}=k_{ye}i_{ye}$, $i_{xi},i_{ye}$ are the currents applied to the coils of the correction motors; $i_{\alpha },i_{\beta }$ are the currents applied to the correction stator coils of the magnetic bearings arranged along the $Ox_{r}$ and $Oy_{r}$ axes to compensate for the angular deviations α and β. The total currents applied to the stator coils in the rotor half-axes *a* and *b* to compensate for the linear deviations $x_{r}$ and $y_{r}$ but also the angular ones α and β are [16] as follows:(3)$$i_{xr}^{a}=i_{x}-i_{\beta },\;i_{xr}^{b}=i_{x}+i_{\beta },\;i_{yr}^{a}=i_{y}+i_{\alpha },\;i_{yr}^{b}=i_{y}-i_{\alpha };$$
the total linear displacements of the rotor semi-axes measured from the center of the mass of the magnetic bearings and those measured by the displacement sensors arranged along the $Ox_{r}$ and $Oy_{r}$ axes are, respectively,$$h_{xr}^{ma}=x_{r}-l_{m}\beta ,\;h_{xr}^{sa}=x_{r}-l_{s}\beta ,\;h_{xr}^{mb}=x_{r}+l_{m}\beta ,\;h_{xr}^{sb}=x_{r}-l_{s}\beta ,\;h_{yr}^{ma}=y_{r}+l_{m}\alpha ,$$(4)$$h_{yr}^{sa}=y_{r}+l_{s}\alpha ,\;h_{yr}^{mb}=y_{r}-l_{m}\alpha ,\;h_{yr}^{sb}=y_{r}-l_{s}\alpha ,$$
where $l_{s}$ is the placement distance of the sensors from the origin of the frame $Ox_{r}y_{r}z_{r}$.

The vector of controlled linear and angular displacements of the gyroscopic rotor $q_{r}$ and, respectively, $q_{s}$ is the vector of total linear displacements (measured by the linear displacement sensors) are (as in [17])(5a)$$q_{r}={\left[\begin{array}{cccc} x_{r} & \beta & y_{r} & \alpha \end{array}\right]}^{T},$$(5b)$$q_{s}={\left[\begin{array}{cccc} h_{xr}^{sa} & h_{xr}^{sb} & h_{yr}^{sa} & h_{yr}^{sb} \end{array}\right]}^{T};$$
and, according to (4),(6)$$q_{r}=C_{s}q_{s}=\frac{1}{2l_{s}}\left[\begin{array}{cccc} l_{s} & l_{s} & 0 & 0\\ -1 & 1 & 0 & 0\\ 0 & 0 & l_{s} & l_{s}\\ 0 & 0 & 1 & -1 \end{array}\right]q_{s}.$$

The dynamics of the gyroscopic gimbals’ rotations are described by the equations presented in [1]:(7a)$${\ddot{\sigma }}_{i}=\frac{J_{iz}-J_{iy}}{J_{ix}}{\dot{\sigma }}_{e}^{2}\sin\sigma_{i}\cos\sigma_{i}+\frac{K_{0}}{J_{ix}}\left({\dot{\sigma }}_{e}\cos\sigma_{i}+\dot{\beta }\right)-\frac{2l_{m}^{2}k_{hy}}{J_{ix}}\alpha -\frac{2l_{m}k_{yr}}{J_{ix}}i_{\alpha }+\frac{M_{xi}}{J_{rx}},$$$${\ddot{\sigma }}_{e}=-\frac{2\left(J_{iz}-J_{iy}\right)}{\lambda_{1}}{\dot{\sigma }}_{i}{\dot{\sigma }}_{e}\sin\sigma_{i}\cos\sigma_{i}-\frac{K_{0}}{\lambda_{1}}\left({\dot{\sigma }}_{i}+\dot{\alpha }\right)\cos\sigma_{i}-\frac{2l_{m}^{2}k_{hx}}{\lambda_{1}}\beta \cos\sigma_{i}-$$(7b)$$-\frac{2l_{m}k_{xr}}{\lambda_{1}}\cos\sigma_{i}i_{\beta }+\frac{M_{ye}}{\lambda_{1}}.$$

Taking into account the fact that the base (flying object A) rotates around its axes (the OXYZ frame) with the angular rates $\omega_{X},\omega_{Y},\omega_{Z}$, it results an angular rate $\omega_{ZT}$, which depends on these $\omega_{ZT}≅\omega_{X}=\dot{\phi }$ is the angular rate of A around its horizontal axis. To compensate the effect of the angular rate, $\omega_{X}$, i.e., to overlap the $Ox_{e}y_{e}z_{e}$ frame over the $Ox_{T}y_{T}z_{T}$ frame, the $Ox_{e}y_{e}z_{e}$ frame should be rotated by the angle ϕ (see Figure 1c). However, the $Oy_{e}-$ axis cannot rotate because the bearings of the outer frame in its axis of rotation are located on the base (the support S is fixed to A). Therefore, it is necessary to make additional rotations around $Oy_{e}$ and $Oy_{i}$ axes. For this, the torque moments generated by the two motors must be functions not only of the angle λ ($\lambda_{i}$ and $\lambda_{e}$ angles, (its components in the two planes perpendicular to the plane of the angle λ)) but also of the angle ϕ. Therefore, according to Figure 1c, the role of the ${\vec{M}}_{ye}-$ moment is taken by ${\vec{M}}_{yT}$, and the role of the ${\vec{M}}_{xi}-$ moment is taken by ${\vec{M}}_{xT}$;(8a)$$M_{yT}=M_{xi}\sin\phi +M_{ye}\cos\phi =k_{xi}\left(\sin\phi \right)i_{xi}+k_{ye}\left(\cos\phi \right)i_{ye},$$(8b)$$M_{xT}=M_{xi}\cos\phi -M_{ye}\sin\phi =k_{xi}\left(\cos\phi \right)i_{xi}-k_{ye}\left(\sin\phi \right)i_{ye}.$$

So, in Equations (2) and (7), $M_{xi}$ and $M_{ye}$ will be replaced by $M_{xT}$ and $M_{yT}$; as a result, the equations systems (1), (2), and (7) can be expressed in the following forms:(9)$${\ddot{y}}_{1}=\left[\begin{array}{cc} \frac{2k_{hx}}{m} & 0\\ 0 & \frac{2k_{hy}}{m} \end{array}\right]y_{1}+\left[\begin{array}{cc} \frac{2k_{xr}}{m} & 0\\ 0 & \frac{2k_{yr}}{m} \end{array}\right]u_{1}+g,$$
where $y_{1}={\left[\begin{array}{cc} x_{r} & y_{r} \end{array}\right]}^{T}$, $u_{1}={\left[\begin{array}{cc} i_{x} & i_{y} \end{array}\right]}^{T}$, $g={\left[\begin{array}{cc} g_{xr} & g_{yr} \end{array}\right]}^{T}$;(10)$${\ddot{y}}_{2}=\left[\begin{array}{c} \begin{array}{c} \frac{J_{rz}}{J_{rx}}\dot{\beta }{\dot{\sigma }}_{e}\sin\sigma_{i}+\frac{J_{iy}+J_{rz}-J_{iz}}{J_{rx}}{\dot{\sigma }}_{e}^{2}\sin\sigma_{i}\cos\sigma_{i}-\frac{K_{0}}{J_{rx}}\left(\dot{\beta }+{\dot{\sigma }}_{e}\cos\sigma_{i}\right)+\frac{4l_{m}^{2}k_{hy}}{J_{rx}}\alpha +\\ +\frac{4l_{m}^{2}k_{yr}}{J_{rx}}i_{\alpha }-\frac{k_{xi}\cos\phi }{J_{rx}}i_{xi}+\frac{k_{ye}\sin\phi }{J_{rx}}i_{ye} \end{array}\\ \begin{array}{c} \frac{2\left(J_{iz}-J_{iy}\right)}{\lambda_{1}}{\dot{\sigma }}_{i}{\dot{\sigma }}_{e}\sin\sigma_{i}\cos^{2}\sigma_{i}+{\dot{\sigma }}_{i}{\dot{\sigma }}_{e}\sin\sigma_{i}-\frac{J_{rz}}{J_{rx}}\left(\dot{\alpha }+{\dot{\sigma }}_{i}\right){\dot{\sigma }}_{e}\sin\sigma_{i}+\frac{K_{0}}{J_{rx}}\left(\dot{\alpha }+{\dot{\sigma }}_{i}\right)+\\ +2l_{m}k_{hr}\frac{\lambda_{1}+J_{rx}\cos^{2}\sigma_{i}}{\lambda_{1}J_{rx}}i_{\beta }+2l_{m}^{2}k_{hx}\frac{\cos^{2}\sigma_{i}}{\lambda_{1}}\beta -\frac{k_{xi}\sin\phi \cos\sigma_{i}}{\lambda_{1}}i_{xi}-\\ -\frac{k_{ye}\cos\phi \cos\sigma_{i}}{\lambda_{1}}i_{ye} \end{array} \end{array}\right],$$
with $y_{2}={\left[\begin{array}{cc} \alpha & \beta \end{array}\right]}^{T}$;(11)$${\ddot{y}}_{3}=\left[\begin{array}{c} \begin{array}{c} \frac{J_{iz}-J_{iy}}{J_{ix}}{\dot{\sigma }}_{e}^{2}\sin\sigma_{i}\cos\sigma_{i}+\frac{K_{0}}{J_{ix}}\left({\dot{\sigma }}_{e}\cos\sigma_{i}+\dot{\beta }\right)-\frac{2l_{m}^{2}k_{hy}}{J_{ix}}\alpha -\frac{2l_{m}k_{yr}}{J_{ix}}i_{\alpha }+\\ +\frac{k_{xi}\cos\phi }{J_{ix}}i_{xi}-\frac{k_{ye}\sin\phi }{J_{ix}}i_{ye} \end{array}\\ \begin{array}{c} \frac{2\left(J_{iy}-J_{iz}\right)}{\lambda_{1}}{\dot{\sigma }}_{i}{\dot{\sigma }}_{e}\sin\sigma_{i}\cos\sigma_{i}-\frac{K_{0}}{\lambda_{1}}\left({\dot{\sigma }}_{i}+\dot{\alpha }\right)\cos\sigma_{i}-\frac{2l_{m}^{2}k_{hx}\cos\sigma_{i}}{\lambda_{1}}i_{\beta }-\\ -\frac{2l_{m}k_{xr}\cos\sigma_{i}}{\lambda_{1}}i_{\beta }+\frac{k_{xi}\sin\phi }{\lambda_{1}}i_{xi}+\frac{k_{ye}\cos\phi }{\lambda_{1}}i_{ye} \end{array} \end{array}\right],$$
with $y_{3}={\left[\begin{array}{cc} \sigma_{i} & \sigma_{e} \end{array}\right]}^{T}$.

## 4. The Design of Adaptive Control Subsystems of DGMSGG’s Dynamics

According to the dynamic models (1), (2), and (7), respectively, (9), (10), and (11), the dynamics of the rotations (precession motion) of the gyroscopic rotor are mutually influenced by the dynamics of the gimbals. The two dynamics can be decoupled and controlled independently, while the influences of the physical couplings between them can be expressed through disturbing vectors. Thus, for the dynamics of the gyroscopic rotor, the disturbance vector $\varepsilon_{2}$ is expressed as a function of the vector $u_{3}$ of the control variables and the vector $y_{3}$ of the output variables (gimbals’ angles $\sigma_{i}$ and $\sigma_{e}$) of the gimbals’ dynamics, while for the gimbals’ dynamics, disturbance vector $\varepsilon_{3}$ is expressed as a function of the vector $u_{2}$ of the control variables and the vector $y_{2}$ of the output variables (precession angles α and β) of the rotor dynamics. In order to compensate the effects of the two disturbances, it is necessary to introduce some adaptive components in the control laws of the two dynamics.

Subsystems (9), (10), and (11) have relative degrees (with respect to the output vectors $y_{1}$, $y_{2}$, and $y_{3}$) $r_{1}=r_{2}=r_{3}=2$.

Let the nonlinear system be described by the following equations:(12a)$$\dot{x}=f\left(x,u\right),$$(12b)$$y=h\left(x\right),$$
where x(n×1) is the state vector, f and h− nonlinear functions ((n×1) and (n×1)), u− input vector, and y− output vector. The system (12) satisfies the conditions of hypothesis 1 in [29](13)$$y^{\left(r\right)}=h_{r}\left(x,u\right),h_{r}\overset{\Delta }{=}\frac{d^{r}y}{dt^{r}}=\frac{d^{r}h}{dt^{r}}=h^{\left(r\right)},\frac{\partial h_{i}}{\partial u}=0,\left(i\in [0,r]\right),\frac{\partial h_{r}}{\partial u}\neq 0,$$
that is, all the derivatives $y^{\left(i\right)},\;\left(i\in [0,r]\right)$ do not depend on *u*, while $y^{\left(r\right)}=h_{r}$ depends on *u*, *r* being the relative degree of the system (12).

For the system described by Equations (12), a control law (pseudo control) is designed after the output vector y, so that $y\left(t\right)$ follows the *r* times derivable $\bar{y}(t)$;(14)$$\hat{v}={\hat{h}}_{r}\left(y,\hat{u}\right),$$
where ${\hat{h}}_{r}\left(y,\hat{u}\right)$ is the best approximation of the function $h_{r}\left(x,u\right)=h_{r}\left(x\left(y\right),u\right)=h_{r}\left(y,u\right)$. The inverses of these functions are(15)$$u=h_{r}^{-1}\left(y,v\right),\;\hat{u}={\hat{h}}_{r}^{-1}\left(y,\hat{v}\right).$$

If ${\hat{h}}_{r}\equiv h_{r}$, then $\;y^{\left(r\right)}=v=\hat{v}$; otherwise,(16)$$y^{\left(r\right)}=v=\hat{v}+\varepsilon ,$$
where(17)$$\varepsilon =h_{r}\left(y,u\right)-{\hat{h}}_{r}\left(y,\hat{u}\right),$$
is the approximation error (of dynamic inversion) of the function $h_{r}\left(y,u\right)$, which behaves as a disturbance for that dynamic ($\varepsilon_{2}$ for the dynamics of the rotor precession, while $\varepsilon_{3}$ for the dynamics of the gyroscopic gimbals).

If the function $u=h_{r}^{-1}\left(y,v\right)$ is developed in Taylor series around $\left(y,\hat{v}\right)$, one obtains, successively,(18)$$u=h_{r}^{-1}\left(y,v\right)\approx {\hat{h}}_{r}^{-1}\left(y,\hat{v}\right)+\frac{d}{dv}{\left(h_{r}^{-1}\left(y,v\right)\right)}_{v=\hat{v}}\left(v-\bar{v}\right)\overset{\left(15\right)}{=}\hat{u}+\frac{d}{dv}\left({\hat{h}}_{r}^{-1}\left(y,\hat{v}\right)\right)\varepsilon .$$

The dynamics (9), (10), and (11) have the forms (12) and fulfill the conditions (13). Therefore, they will be represented in the form (16). In Equation (9), the term −g plays the role of $\varepsilon_{1}$; it results in ${\ddot{y}}_{1}=v_{1}$, $v_{1}={\hat{v}}_{1}+\varepsilon_{1}={\hat{v}}_{1}-g$.(19)$${\hat{v}}_{1}={\hat{h}}_{r1}\left(y_{1},{\hat{u}}_{1}\right)=\left[\begin{array}{cc} \frac{2k_{hx}}{m} & 0\\ 0 & \frac{2k_{hy}}{m} \end{array}\right]y_{1}+\left[\begin{array}{cc} \frac{2k_{xr}}{m} & 0\\ 0 & \frac{2k_{yr}}{m} \end{array}\right]{\hat{u}}_{1}.$$

The inverse function is(20)$${\hat{u}}_{1}={\hat{h}}_{r1}^{-1}\left(y_{1},{\hat{v}}_{1}\right)={\left[\begin{array}{cc} \frac{2k_{xr}}{m} & 0\\ 0 & \frac{2k_{yr}}{m} \end{array}\right]}^{-1}\left\{{\hat{v}}_{1}-\left[\begin{array}{cc} \frac{2k_{hx}}{m} & 0\\ 0 & \frac{2k_{hy}}{m} \end{array}\right]\begin{array}{c} y_{1} \end{array}\right\},$$
and, according to (18), one obtains(21)$$u_{1}={\hat{h}}_{r1}^{-1}\left(y_{1},v_{1}\right)={\left[\begin{array}{cc} \frac{2k_{hx}}{m} & 0\\ 0 & \frac{2k_{hy}}{m} \end{array}\right]}^{-1}\left\{v_{1}-\left[\begin{array}{cc} \frac{2k_{hx}}{m} & 0\\ 0 & \frac{2k_{hy}}{m} \end{array}\right]\begin{array}{c} y_{1} \end{array}\right\}.$$

The dynamics (10) may be described by the equation ${\ddot{y}}_{2}={\hat{v}}_{2}+\varepsilon_{2}$, with ${\hat{u}}_{2}={\left[\begin{array}{cc} {\hat{i}}_{\alpha } & {\hat{i}}_{\beta } \end{array}\right]}^{T}$,(22)$${\hat{v}}_{2}={\hat{h}}_{r2}\left(y_{2},{\hat{u}}_{2}\right)=\left[\begin{array}{cc} \frac{4l_{m}^{2}k_{hy}}{J_{rx}} & 0\\ 0 & \frac{2l_{m}^{2}k_{hx}}{J_{rx}} \end{array}\right]y_{2}+\left[\begin{array}{cc} \frac{4l_{m}^{2}k_{yr}}{J_{rx}} & 0\\ 0 & \frac{2l_{m}^{2}k_{xr}}{J_{rx}} \end{array}\right]{\hat{u}}_{2},$$(23)$$\varepsilon_{2}=\left[\begin{array}{c} \begin{array}{c} \frac{J_{rz}}{J_{rx}}\hat{\dot{\beta }}{\hat{\dot{\sigma }}}_{e}\sin\sigma_{i}+\frac{J_{iy}+J_{rz}-J_{iz}}{J_{rx}}{\hat{\dot{\sigma }}}_{e}^{2}\sin\sigma_{i}\cos\sigma_{i}-\frac{K_{0}}{J_{rx}}\left(\hat{\dot{\beta }}+{\hat{\dot{\sigma }}}_{e}\cos\sigma_{i}\right)-\\ -\frac{k_{xi}\cos\phi }{J_{rx}}i_{xi}+\frac{k_{ye}\sin\phi }{J_{rx}}i_{ye} \end{array}\\ \begin{array}{c} \frac{2\left(J_{iz}-J_{iy}\right)}{\lambda_{1}}{\hat{\dot{\sigma }}}_{i}{\hat{\dot{\sigma }}}_{e}\sin\sigma_{i}\cos^{2}\sigma_{i}+{\hat{\dot{\sigma }}}_{i}{\hat{\dot{\sigma }}}_{e}\sin\sigma_{i}-\frac{J_{rz}}{J_{rx}}\left(\hat{\dot{\alpha }}+{\hat{\dot{\sigma }}}_{i}\right){\hat{\dot{\sigma }}}_{e}\sin\sigma_{i}+\frac{K_{0}}{J_{rx}}\left(\hat{\dot{\alpha }}+{\hat{\dot{\sigma }}}_{i}\right)+\\ +\frac{2l_{m}k_{xr}\cos^{2}\sigma_{i}}{\lambda_{1}}i_{\beta }-\frac{k_{xi}\sin\phi \cos\sigma_{i}}{\lambda_{1}}i_{xi}-\frac{k_{ye}\cos\phi \cos\sigma_{i}}{\lambda_{1}}i_{ye} \end{array} \end{array}\right];$$
in the above-presented equations, the variables with “^” represent the estimated values of those variables (see Equations (57) and (58)).

The vector $\varepsilon_{2}$ contains all the terms in Equation (10), which are functions of $x_{2}-$ vector’s state variables (other than output $y_{2}-$ vector’s components), as well as of the command variables, $u_{3}-$ vector’s components.

From (22) it results in(24)$${\hat{u}}_{2}=\left[\begin{array}{c} {\hat{i}}_{\alpha }\\ {\hat{i}}_{\beta } \end{array}\right]={\hat{h}}_{r2}^{-1}\left(y_{2},{\hat{v}}_{2}\right)={\left[\begin{array}{cc} \frac{4l_{m}k_{yr}}{J_{rx}} & 0\\ 0 & \frac{2l_{m}k_{xr}}{J_{rx}} \end{array}\right]}^{-1}+\left\{{\hat{v}}_{2}-\left[\begin{array}{cc} \frac{4l_{m}^{2}k_{hy}}{J_{rx}} & 0\\ 0 & \frac{2l_{m}^{2}k_{hx}}{J_{rx}} \end{array}\right]\begin{array}{c} \begin{array}{l} y_{2} \end{array} \end{array}\right\}.$$

Introducing (24) into (18), one obtains(25)$$u_{2}=\left[\begin{array}{c} i_{\alpha }\\ i_{\beta } \end{array}\right]={\hat{h}}_{r2}^{-1}\left(y_{2},v_{2}\right)={\left[\begin{array}{cc} \frac{4l_{m}k_{yr}}{J_{rx}} & 0\\ 0 & \frac{2l_{m}k_{xr}}{J_{rx}} \end{array}\right]}^{-1}+\left\{v_{2}-\left[\begin{array}{cc} \frac{4l_{m}^{2}k_{hy}}{J_{rx}} & 0\\ 0 & \frac{2l_{m}^{2}k_{hx}}{J_{rx}} \end{array}\right]\begin{array}{c} \begin{array}{l} y_{2} \end{array} \end{array}\right\}.$$

The (11)—dynamics is described by the equation ${\ddot{y}}_{3}={\hat{v}}_{3}+\varepsilon_{3}$, with ${\hat{u}}_{3}={\left[\begin{array}{cc} {\hat{i}}_{x_{i}} & {\hat{i}}_{y_{e}} \end{array}\right]}^{T}$,(26)$${\hat{v}}_{3}=\left[\begin{array}{cc} \frac{k_{xi}\cos\phi }{J_{ix}} & -\frac{k_{ye}\sin\phi }{J_{ix}}\\ \frac{k_{xi}\sin\phi }{\lambda_{1}} & \frac{k_{ye}\cos\phi }{\lambda_{1}} \end{array}\right]{\hat{u}}_{3};$$(27)$$\varepsilon_{3}=\left[\begin{array}{c} \begin{array}{c} \frac{J_{iz}-J_{iy}}{J_{ix}}{\hat{\dot{\sigma }}}_{e}^{2}\sin\sigma_{i}\cos\sigma_{i}+\frac{K_{0}}{J_{ix}}{\hat{\dot{\sigma }}}_{e}\cos\sigma_{i}-\frac{2l_{m}^{2}k_{hy}}{J_{ix}}\alpha -\frac{2l_{m}k_{yr}}{J_{ix}}i_{\alpha }+\frac{K_{0}}{J_{ix}}\hat{\dot{\beta }} \end{array}\\ \begin{array}{c} \frac{2\left(J_{iy}-J_{iz}\right)}{\lambda_{1}}{\hat{\dot{\sigma }}}_{i}{\hat{\dot{\sigma }}}_{e}\sin\sigma_{i}\cos\sigma_{i}-\frac{K_{0}}{\lambda_{1}}{\hat{\dot{\sigma }}}_{i}\cos\sigma_{i}-\frac{2l_{m}^{2}k_{hx}\cos\sigma_{i}}{\lambda_{1}}\beta -\\ -\frac{2l_{m}k_{xr}\cos\sigma_{i}}{\lambda_{1}}i_{\beta }-\frac{K_{0}}{\lambda_{1}}\hat{\dot{\alpha }}\cos\sigma_{i} \end{array} \end{array}\right].$$

From (26) it results in(28)$${\hat{u}}_{3}=\left[\begin{array}{c} {\hat{i}}_{xi}\\ {\hat{i}}_{yi} \end{array}\right]={\hat{h}}_{r3}^{-1}\left(y_{3},{\hat{v}}_{3}\right)={\left[\begin{array}{cc} \frac{k_{xi}\cos\phi }{J_{ix}} & -\frac{k_{ye}\sin\phi }{J_{ix}}\\ \frac{k_{xi}\sin\phi }{\lambda_{1}} & \frac{k_{ye}\cos\phi }{\lambda_{1}} \end{array}\right]}^{-1}{\hat{v}}_{3},$$
and according to Equation (18),(29)$$u_{3}=\left[\begin{array}{c} i_{xi}\\ i_{yi} \end{array}\right]={\hat{h}}_{r3}^{-1}\left(y_{3},v_{3}\right)={\left[\begin{array}{cc} \frac{k_{xi}\cos\phi }{J_{ix}} & -\frac{k_{ye}\sin\phi }{J_{ix}}\\ \frac{k_{xi}\sin\phi }{\lambda_{1}} & \frac{k_{ye}\cos\phi }{\lambda_{1}} \end{array}\right]}^{-1}v_{3},$$

Therefore, for each of Equations (9)–(11), the conditions of hypothesis 1 from [29] are used, which means Equations (12) and (13); it results in the relative degree *r* of the subsystem *j* in relation to the output $y_{j}$ and to the input $u_{j}(y=y_{j};\;u=u_{j};\;j=\bar{1,3})$.

If the dynamics described by equation $\ddot{y}=h_{r}(y,u)=v$ have one of the forms (9) to (11), the function
${\hat{h}}_{r}(y,u)=\hat{v}$ is selected from it; that is, the function containing only the output vector y and the input vector u; the remaining terms in $h_{r}$ are introduced into the vector ε, which is the approximation error vector (see Equations (14)–(18)). The function describing the inverse dynamics (the inverse of the $h_{r}$ function) is $u={\hat{h}}_{r}\left(y,v\right)$. Thus, Formulas (20)–(28) were obtained.

Considering the fact that the relative degree in relation to each of the output variables is r=2, for each of the three structures in Figure 2 and Figure 3, one reference model of 2nd order is chosen each with the transfer matrix form from [32].(30)$$H_{m}\left(s\right)=\frac{\omega_{r0}^{2}}{s^{2}+2\xi_{0}\omega_{r0}s+\omega_{r0}^{2}}I_{2},$$
where $I_{2}-$ the unit matrix (2 × 2), $\xi_{0}=0.7$ and $\omega_{r0}=2.5$ rad/s.

The mission of the adaptive component $v_{aj}\left(j=\bar{2,3}\right)$ is to compensate the inversion error $\varepsilon_{j}$; for a stabilized regime, $v_{aj}=\varepsilon_{j}=0$, and the dynamic compensator’s output $v_{j_{cd}}=0$; implicitly, $h_{rj}{\hat{h}}_{rg}=I_{2}$ and ${\ddot{y}}_{j}=v_{j}={\hat{v}}_{j}\equiv {\hat{v}}_{rj}$. The presence on the direct path of each system in Figure 2 and Figure 3 of some 2nd order ideal integrators, and the choice of the P.D. type or P.I.D. type linear dynamic compensators, leads to the conclusion that, in a stabilized regime, $y_{j}={\bar{y}}_{j}$ and ${\dot{y}}_{j}={\dot{\bar{y}}}_{j},\left(j=\bar{2,3}\right)$. If one chooses ${\hat{v}}_{rj}={\ddot{\bar{y}}}_{j}$, then, for a stabilized mode, ${\ddot{y}}_{j}={\hat{v}}_{rj}={\ddot{\bar{y}}}_{j}$; this is why the component ${\hat{v}}_{rj}={\ddot{\bar{y}}}_{j}$ was introduced into ${\hat{v}}_{j}$.

In order to increase the accuracy of the control system of the gyroscopic rotor’s linear deviations $x_{r}$ and $y_{r}$ compared to the zero values imposed on the control law of the vector $y_{1}$, an integrating component was also introduced. So, for the structure in Figure 2a, a dynamic compensator is chosen, whose transfer matrix is(31)$$H_{cd}\left(s\right)=\left(K_{p1}+\frac{K_{i1}}{s}+K_{d1}s\right)I_{2},$$
where $K_{p1}=k_{p1}I_{2}$, $K_{i1}=k_{i1}I_{2}$, $K_{d1}=k_{d1}I_{2}$, while for the systems in Figure 2b and Figure 3(32)$$H_{cd}\left(s\right)=\left(K_{pj}+K_{dj}s\right)I_{2},j=\bar{2,3},$$
where $K_{pj}=k_{pj}I_{2}$, $K_{dj}=k_{dj}I_{2}$.

To calculate the coefficients of the linear dynamic compensators, the following characteristic equation is used:(33)$$I_{2}+H_{cd}\left(s\right)\frac{1}{s^{2}}I_{2}=0,$$
based on the hypothesis $h_{rj}{\hat{h}}_{rj}^{-1}≅I_{2},\;j=\bar{1,3}$.

Equation (33) is equivalent, for the systems in Figure 2a, to the following form:(34)$$s^{3}+k_{d1}s^{2}+k_{p1}s+k_{i1}=0.$$

The output of the P.I.D. type dynamic compensator in Figure 2a is(35)$$v_{1pid}=K_{i1}\int {\tilde{y}}_{1}+K_{p1}{\tilde{y}}_{1}+K_{d1}{\dot{\tilde{y}}}_{1},$$
where $K_{i1}=k_{i1}I_{2}$, $K_{p1}=k_{p1}I_{2}$, $K_{d1}=k_{d1}I_{2}$, ${\tilde{y}}_{1}={\bar{y}}_{1}-y_{1}=\left[\begin{array}{cc} {\bar{x}}_{r}-x_{r} & {\dot{\bar{x}}}_{r}-{\dot{x}}_{r} \end{array}\right]$.

According to Figure 2a,(36)$$v_{1}={\hat{v}}_{r1}+v_{1pid}-g={\ddot{y}}_{1}+K_{i1}\int {\tilde{y}}_{1}+K_{p1}{\tilde{y}}_{1}+K_{d1}{\dot{\tilde{y}}}_{1}-g.$$

Introducing Equation (36) into ${\ddot{y}}_{1}=v_{1}$, one obtains a new form, as follows:(37)$$y_{1}={\ddot{\tilde{y}}}_{1}+K_{i1}\int {\tilde{y}}_{1}+K_{p1}{\tilde{y}}_{1}+K_{d1}{\dot{\tilde{y}}}_{1}-g,$$
equivalent to(38)$${\ddot{\tilde{y}}}_{1}=-K_{i1}\int {\tilde{y}}_{1}-K_{p1}{\tilde{y}}_{1}-K_{d1}{\dot{\tilde{y}}}_{1}+g.$$

Let E be the state vector of the linear subsystem resulting from the compensation of the $h_{r1}-$ function with the ${\hat{h}}_{r1}^{-1}-$ function $\left({\hat{h}}_{r1}^{-1}\equiv h_{r1}^{-1}\right)$; $E=\left[\begin{array}{ccc} E_{1}^{T} & E_{2}^{T} & E_{3}^{T} \end{array}\right]$, where(39)$$E_{1}=\int {\tilde{y}}_{1}={\left[\begin{array}{cc} \int {\tilde{y}}_{11} & \int {\tilde{y}}_{12} \end{array}\right]}^{T},E_{2}={\tilde{y}}_{1}={\left[\begin{array}{cc} {\tilde{y}}_{11} & {\tilde{y}}_{12} \end{array}\right]}^{T},E_{3}={\dot{\tilde{y}}}_{1}={\left[\begin{array}{cc} {\dot{\tilde{y}}}_{11} & {\dot{\tilde{y}}}_{12} \end{array}\right]}^{T}.$$

Equation (38) is equivalent to the following state equation system:(40a)$${\dot{E}}_{1}=E_{2},$$(40b)$${\dot{E}}_{2}=E_{3},$$(40c)$${\dot{E}}_{3}=-K_{i1}E_{1}-K_{p1}E_{2}-K_{d1}E_{3}+g,$$
or to the state equation(40d)$$\dot{E}=A_{1}E+B_{1}g,$$
where the matrices $A_{1}\left(6\times 6\right)$ and $B_{1}\left(6\times 2\right)$ have the forms(41)$$A_{1}=\left[\begin{array}{ccc} 0_{2\times 2} & I_{2} & 0_{2\times 2}\\ 0_{2\times 2} & 0_{2\times 2} & I_{2}\\ -K_{i1} & -K_{p1} & -K_{d1} \end{array}\right],\;B_{1}=\left[\begin{array}{c} 0_{2\times 2}\\ 0_{2\times 2}\\ I_{2} \end{array}\right].$$

The linear state observer is described by the following equation (where $\hat{E}$ is the estimate of the state vector E):(42)$$\dot{\hat{E}}=A_{1}\hat{E}+L_{1}\left({\tilde{y}}_{1}-{\hat{\tilde{y}}}_{1}\right),$$
where (43a)$${\tilde{y}}_{1}=E_{1}=C_{1}E,$$(43b)$${\hat{\tilde{y}}}_{1}={\hat{E}}_{2}=C_{1}\hat{E},$$(43c)$$C_{1}=\left[\begin{array}{ccc} 0_{2\times 2} & I_{2} & 0_{2\times 2} \end{array}\right].$$

State observer’s Equation (42) is equivalent to(44a)$$\dot{\hat{E}}={\bar{A}}_{1}\hat{E}+L_{1}{\tilde{y}}_{1},$$
where(44b)$${\bar{A}}_{1}=A_{1}-L_{1}C_{1}.$$

Linear state observer’s amplification matrix $L_{1}$ is calculated so that the matrix ${\bar{A}}_{1}$ has the imposed eigenvalues located in the left complex semiplane.

The coefficients $k_{i1},\;k_{p1}$, and $k_{d1}$ of the matrices $K_{i1},\;K_{p1}$, and $K_{d1}$ are calculated according to the roots imposed on Equation (34).

According to Equation (35), the command law $v_{1pid}$ might be expressed by one of the following forms:(45a)$$v_{1pid}=D_{c1}E,$$(45b)$${\hat{v}}_{1pid}=D_{c1}\hat{E},$$
where(45c)$$D_{c1}={\left[\begin{array}{ccc} K_{i1} & K_{p1} & K_{d1} \end{array}\right]}^{T};$$
the second form, (45b), is advantageous, because the first form, (45a), involves the introduction of additional sensors (linear speed sensors, for ${\dot{x}}_{r}$ and ${\dot{y}}_{r}$) to determine the derivative component $K_{d1}{\dot{\tilde{y}}}_{1}=K_{d1}\left({\dot{\bar{y}}}_{1}-{\dot{y}}_{1}\right)$, while the second form uses the estimated vector $\hat{E}=\left[\begin{array}{ccc} {\hat{E}}_{1}^{T} & {\hat{E}}_{2}^{T} & {\hat{E}}_{3}^{T} \end{array}\right]$.

For the subsystem in Figure 2b and the stabilizing subsystem in Figure 3,(46)$$v_{jpd}=K_{pj}{\tilde{y}}_{2}+K_{dj}{\dot{\tilde{y}}}_{2},\;j=\bar{2,3},$$
with $K_{p2}=k_{p2}I_{2}$ and $K_{d2}=k_{d2}I_{2}$.

The command law $v_{j}$ has the form(47)$$v_{j}={\hat{v}}_{2}+\varepsilon_{j}={\hat{v}}_{rj}+{\hat{v}}_{jpd}-v_{aj}+\varepsilon_{j}={\ddot{\bar{y}}}_{j}+K_{pj}{\tilde{y}}_{j}+K_{dj}{\dot{\tilde{y}}}_{j}-v_{aj}+\varepsilon_{j};\;j=\bar{2,3}\;.$$

The state vector of each linear system *j* ($j=\bar{2,3}$) consisting of a linear dynamic compensator and the subsystem with the transfer matrix $H_{dj}\left(s\right)=\frac{1}{s^{2}}I_{2}$ is(48)$$E_{j}^{/}={\left[\begin{array}{cc} E_{1j}^{{/}^{T}} & E_{2j}^{{/}^{T}} \end{array}\right]}^{T}={\left[\begin{array}{cc} {\tilde{y}}_{j}^{T} & {\dot{\tilde{y}}}_{j}^{T} \end{array}\right]}^{T},j=\bar{2,3};$$
for j=2, $E_{2}^{/}={\left[\begin{array}{cc} E_{12}^{{/}^{T}} & E_{22}^{{/}^{T}} \end{array}\right]}^{T}={\left[\begin{array}{cc} e_{1}^{T} & e_{2}^{T} \end{array}\right]}^{T}={\left[\begin{array}{cc} {\tilde{y}}_{2}^{T} & {\dot{\tilde{y}}}_{2}^{T} \end{array}\right]}^{T}$, while for j=3 one obtains $E_{3}^{/}={\left[\begin{array}{cc} E_{13}^{{/}^{T}} & E_{23}^{{/}^{T}} \end{array}\right]}^{T}={\left[\begin{array}{cc} e_{1}^{{/}^{T}} & e_{2}^{{/}^{T}} \end{array}\right]}^{T}={\left[\begin{array}{cc} {\tilde{y}}_{3}^{T} & {\dot{\tilde{y}}}_{3}^{T} \end{array}\right]}^{T}$.

Introducing Equation (47) into ${\ddot{y}}_{j}=v_{j}$, one obtains the equation of the linear system with the input $v_{aj}-\varepsilon_{j}$ and the output ${\tilde{y}}_{j}$:(49a)$${\ddot{\tilde{y}}}_{j}=-K_{pj}{\tilde{y}}_{j}-K_{dj}{\dot{\tilde{y}}}_{j}+\left(v_{aj}-\varepsilon_{j}\right);\;j=\bar{2,3},$$
equation equivalent to the following state equation system(49b)$${\dot{E}}_{1j}^{/}=E_{2j}^{/},$$(49c)$${\dot{E}}_{2j}^{/}=-K_{pj}E_{1j}^{/}-K_{dj}E_{2j}^{/}+\left(v_{aj}-\varepsilon_{j}\right);\;j=\bar{2,3},$$
respectively, to the state equation(50)$${\dot{E}}_{j}^{/}=A_{j}E_{j}^{/}+B_{j}\left(v_{aj}-\varepsilon_{j}\right),$$
where(51)$$A_{j}={\left[\begin{array}{cc} 0_{2\times 2} & I_{2}\\ -K_{pj} & -K_{dj} \end{array}\right]}_{\left(4\times 4\right)},B_{j}={\left[\begin{array}{c} 0_{2\times 2}\\ I_{2} \end{array}\right]}_{\left(4\times 2\right)}.$$

The linear state observer (with ${\hat{E}}_{j}^{/}-$ the estimate of the vector $E_{j}^{/}$) is described by the following equations:(52a)$${\dot{\hat{E}}}_{j}^{/}=A_{j}{\hat{E}}_{j}^{/}+L_{j}\left({\tilde{y}}_{j}-{\hat{\tilde{y}}}_{j}\right),$$(52b)$${\tilde{y}}_{j}=E_{1j}^{/}=C_{j}E_{j}^{/}{\hat{\tilde{y}}}_{j}={\hat{E}}_{1j}^{/}=C_{j}{\tilde{E}}_{j}^{/},$$
with(53)$$C_{j}={\left[\begin{array}{cc} I_{2} & 0_{2\times 2} \end{array}\right]}_{\left(2\times 4\right)}.$$

The Equation (52) are equivalent to(54a)$${\dot{\hat{E}}}_{j}^{/}={\bar{A}}_{j}{\hat{E}}_{j}^{/}+L_{j}{\tilde{y}}_{j},$$(54b)$${\bar{A}}_{j}=A_{j}-L_{j}C_{j};\;j=\bar{2,3}.$$

The amplification matrix $L_{j}$ of the linear state observer *j* is calculated so that the matrix ${\bar{A}}_{j}$ has the imposed eigenvalues located in the left complex semiplane.

According to Equation (46), the command law $v_{jpd}$ might be expressed by one of the following forms:(55a)$$v_{jpd}=D_{cj}E_{j}^{/},$$(55b)$${\hat{v}}_{jpd}=D_{cj}{\hat{E}}_{j}^{/},$$
where(55c)$$D_{cj}={\left[\begin{array}{cc} K_{pj} & K_{dj} \end{array}\right]}^{T};j=\bar{2,3}.$$

To avoid the need to introduce additional sensors for angular velocities ($\dot{\alpha }$ and $\dot{\beta }$, respectively, ${\dot{\sigma }}_{i}$ and ${\dot{\sigma }}_{e}$), the second form of the law (55b) will be used.

The coefficients $k_{pj}$ and $k_{dj}$ of the matrices $K_{pj}$ and $K_{dj}$ are calculated according to the roots imposed on the following equation:(56)$$s^{2}+k_{dj}s+k_{pj}=0,\;j=\bar{2,3}.$$

We specify the fact that in the calculation form of the function ${\hat{h}}_{r3}^{-1}$, of the form in (29), respectively, in Equation (23) for the calculation of $\varepsilon_{2}$, the estimated vectors ${\hat{\dot{y}}}_{2}={\left[\begin{array}{cc} \hat{\dot{\alpha }} & \hat{\dot{\beta }} \end{array}\right]}^{T}$ and ${\hat{\dot{y}}}_{3}={\left[\begin{array}{cc} {\hat{\dot{\sigma }}}_{i} & {\hat{\dot{\sigma }}}_{e} \end{array}\right]}^{T}$ are used instead of the vectors ${\dot{y}}_{2}={\left[\begin{array}{cc} \dot{\alpha } & \dot{\beta } \end{array}\right]}^{T}$ and ${\dot{y}}_{3}={\left[\begin{array}{cc} {\dot{\sigma }}_{i} & {\dot{\sigma }}_{e} \end{array}\right]}^{T}$. Therefore,(57)$${\hat{\dot{y}}}_{2}={\dot{\bar{y}}}_{2}-{\hat{\dot{\tilde{y}}}}_{2}={\dot{\bar{y}}}_{2}-{\hat{e}}_{2}={\dot{\bar{y}}}_{2}-\left(\mathrm{Mat}\_1\right)e,{\hat{\dot{y}}}_{2}={\left[\begin{array}{cc} \hat{\dot{\alpha }} & \hat{\dot{\beta }} \end{array}\right]}^{T},$$
with ${\hat{e}}_{2}$ as the second component of the estimated vector $\hat{e}$, respectively,(58)$${\hat{\dot{y}}}_{3}={\dot{\bar{y}}}_{3}-{\hat{\dot{\tilde{y}}}}_{3}={\dot{\bar{y}}}_{3}-{\hat{e}}_{2}^{/}={\dot{\bar{y}}}_{3}-\left(\mathrm{Mat}\_1\right)e^{/},{\hat{\dot{y}}}_{3}={\left[\begin{array}{cc} {\dot{\sigma }}_{i} & {\dot{\sigma }}_{e} \end{array}\right]}^{T},$$
with ${\hat{e}}_{2}^{/}$ as the second component of the estimated vector ${\hat{e}}^{/}$; ${\dot{\bar{y}}}_{2}$ and ${\dot{\bar{y}}}_{3}$ are provided by the reference models; $\mathrm{Mat}\_1=\left[\begin{array}{cc} 0_{2\times 2} & I_{2} \end{array}\right]$. Similarly, ${\hat{y}}_{2}={\bar{y}}_{2}-{\hat{e}}_{1}={\bar{y}}_{2}-\left(\mathrm{Mat}\_12\right)\hat{e}$ and ${\hat{y}}_{3}={\bar{y}}_{3}-\left(\mathrm{Mat}\_12\right){\hat{e}}^{/}$; $\mathrm{Mat}\_12=\left[\begin{array}{cc} I_{2} & 0_{2\times 2} \end{array}\right]$.

With the estimated vectors $\hat{e}$ and ${\hat{e}}^{/}$, the $NN_{c}-$ neural network’s training vectors are calculated:(59)$$\bar{e}={\hat{e}}^{T}P_{2}B_{2},\;{\bar{e}}^{/}={{\hat{e}}^{/}}^{T}P_{3}B_{3},$$
where $P_{2}$ and $P_{3}$ are $\left(4\times 4\right)$ matrices, solutions of the Lyapunov equations:(60)$${\bar{A}}_{2}^{T}P_{2}+P_{2}{\bar{A}}_{2}=-Q_{2},\;{\bar{A}}_{3}^{T}P_{3}+P_{3}{\bar{A}}_{3}=-Q_{3};$$
$Q_{2}$ and $Q_{3}$ are positively defined matrices, while the adaptive command laws are calculated with the formulas in [30]:(61)$$v_{aj}=W_{j}^{T}\sigma_{j}\left(V_{j}\eta_{j}\right),j=\bar{2,3},$$
where $W_{j}$ and $V_{j}$ are the weight matrices of the neural networks $NN_{cj}$, the solutions of the differential equations(62a)$${\dot{W}}_{j}=-\Gamma_{W}\left[2\left(\sigma_{j}-\sigma_{j}^{/}V_{j}^{T}\eta_{j}\right){\bar{e}}_{j}+k_{2}\left(W_{j}-W_{j0}\right)\right],\;W_{j0}=W_{j}\left(0\right),$$(62b)$${\dot{V}}_{j}=-\Gamma_{V}\left[2\eta_{j}{\bar{e}}_{j}W_{j}^{T}\sigma_{j}^{/}+k_{2}\left(V_{j}-V_{j0}\right)\right],\;V_{j0}=V_{j}\left(0\right),j=\bar{2,3},$$
with ${\bar{e}}_{2}=\bar{e}$ and ${\bar{e}}_{3}={\bar{e}}^{/}$, $\sigma^{/}-$ the derivative of the sigmoidal function $\sigma_{j}\left(z\right)=\sigma_{j}\left(V_{j}^{T}\eta_{j}\right)$ having the form(63)$$\sigma_{j}\left(z\right)=\frac{1}{1+e^{-az}},a^{T}={\left[\begin{array}{ccc} a_{1} & \cdots & a_{n_{ij}+1} \end{array}\right]}_{\left(1\times n_{2j}\right)},j=\bar{2,3},$$
$a_{i},i=\bar{1,n_{ij}+1}$, being the activation potentials, $n_{1j}$ and $n_{2j}-$ the number of neurons in the input layer and, respectively, in the hidden layer of the neural network *j*.

The input vector $\eta_{j},\;j=\bar{2,3}$, of the $NN_{j}-$ neural network has the form in [30]:(64)$$\eta_{j}={\left[1\;{\hat{v}}_{jd}^{T}\;v_{jd}^{T}\right]}^{T}={\left[1\;v_{j}^{T}\left(t\right)\;v_{j}^{T}\left(t-d\right)y_{j}\left(t\right)\;y_{j}\left(t-d\right)\;......\;y_{j}\left[t-\left(n_{1j}-1\right)d\right]\right]}^{T},$$
where d is the sampling step.

The AMB rotor’s translation dynamics are physically decoupled from the other two dynamics and, implicitly, the control of the first one is decoupled from the control of the other two. These ones, the dynamics 2 (of the AMB rotor’s rotation) and the dynamics 3 (of the gyroscopic gimbal’s rotations) are connected (intrinsically, physically, and phenomenologically). These two dynamics’ terms, other than those that contain exclusively the dynamics’ inputs and outputs, are included in the blocks for calculating dynamic inversion errors ($\varepsilon_{2}$ and $\varepsilon_{3}$); the effects of these dynamic inversion errors are compensated by the adaptive components of the control laws ($v_{aj}\left(j=\bar{2,3}\right)$) provided by neural networks. As a result, in a stabilized regime, the two subsystems effectively become (physically) decoupled.

The adaptive control law for the compensation of the disturbance effect was deduced by the authors of the paper [30], using the theory of Lyapunov functions, for a wide class of nonlinear systems that verifies the conditions of hypothesis 1 in [29].

According to Figure 1a,c, the elevations of the guide line $Oz_{T}$ and of the sight line $Oz_{i}$, as well as the deviation of the sight line from the guide line, can be expressed in the form of algebraic vectors, which have as component elements the respective angles in the planes $Oy_{i}z_{i}$ and $Oz_{i}x_{i}$,(65)$$\sigma_{t}={\left[\begin{array}{cc} \sigma_{ti} & \sigma_{te} \end{array}\right]}^{T}=y_{3c},\sigma ={\left[\begin{array}{cc} \sigma_{i} & \sigma_{e} \end{array}\right]}^{T}=y_{3},\lambda ={\left[\begin{array}{cc} \lambda_{i} & \lambda_{e} \end{array}\right]}^{T}.$$

In stabilized mode, the gyroscopic rotor is centered, which means that the gyroscopic rotor’s $Oz_{r}-$ axis overlaps the inner gimbal’s $Oz_{i}-$ axis (identical to the CT’s axis); therefore,(66)$$x_{r}=y_{r}=0,\;\alpha =\beta =0,$$
while the line of sight overlaps the target line (the guide line), that means that(67)$$\sigma =\sigma_{t},\lambda =0,$$
and the kinetic moment vector $K_{0}$ is oriented in the direction of the guide line.

The guidance controller will be chosen as P.I. type, having the output(68)$$v_{3pi}=K_{i0}\int {\tilde{y}}_{3\sigma }+K_{p0}{\tilde{y}}_{3\sigma },$$
with $K_{i0}=k_{i0}I_{2}$ and $K_{p0}=k_{p0}I_{2}$.

Figure 3b depicts the block diagram of the linear subsystem (for $v_{3}=\varepsilon_{3}$, ${\hat{h}}_{r3}^{-1}h_{r3}\approx I_{2}$). The inner (stabilization) contour has the transfer matrix(69)$$H_{s}\left(s\right)=\frac{1}{s^{2}+k_{d3}s+k_{p3}}I_{1},$$
while the outer (guidance, orientation) contour has the transfer matrix(70)$$H_{s}\left(s\right)=\frac{k_{po}s+k_{io}}{s^{3}+k_{d3}s^{2}+\left(k_{po}+k_{p3}\right)s+k_{io}}I_{4}.$$

For the calculation of the coefficients $k_{po}$ and $k_{io}$ of the matrices $K_{po}$ and $K_{io}$, the roots of the characteristic equation(71)$$s^{3}+k_{d3}s^{2}+\left(k_{po}+k_{p3}\right)s+k_{io}=0,$$
must be located in the left complex semiplane.

According to Figure 3b,(72)$$\lambda =-\frac{I_{2}}{I_{2}+H_{i}\left(s\right)H_{s}\left(s\right)}\sigma_{t}=-\frac{s\left(s^{2}+k_{d3}s+k_{p3}\right)}{s^{3}+k_{d3}s^{2}+\left(k_{po}+k_{p3}\right)s+k_{io}}\sigma_{t}=-\frac{s^{2}+k_{d3}s+k_{p3}}{s^{3}+k_{d3}s^{2}+\left(k_{po}+k_{p3}\right)s+k_{io}}{\dot{\sigma }}_{t},$$
in stabilized mode(73)$$\lambda =-\frac{k_{p3}}{k_{io}}{\dot{\sigma }}_{t},$$
so λ**~**${\dot{\sigma }}_{t}=\omega_{t}$ (OT—guidance line’s angular rate). Therefore, the signals proportional to the λ− vector’s components $\lambda_{i}$ and $\lambda_{e}$, provided by the CT target’s coordinator, are applied to the flight vehicle’s (missile) autopilot, in order to orientate it toward the T—target, so, the guidance line translates parallel to itself until the interception point (${\dot{\sigma }}_{t}={\left[\begin{array}{cc} 0 & 0 \end{array}\right]}^{T}$), according to the self-steering method by parallel approach [31].

## 5. Numerical Simulations

One has studied the dynamics of the DGMSGG, consisting of the systems shown in Figure 2. The numerical values of the parameters used for the calculations are as follows: m=2.8 kg; $l_{m}=4.1\times 10^{-2}$ m; $l_{s}=6.5\times 10^{-2}$ m; $k_{xr}=k_{yr}=0.21$ N/A; $k_{hx}=k_{hy}=8$ N/m; $J_{rx}=J_{ry}=5\times 10^{-2}$ N.m.s^2^/rad; $J_{rz}=6\times 10^{-2}$ N.m.s^2^/rad; $J_{ix}=J_{iy}=5\times 10^{-2}$ N.m.s^2^/rad; $J_{iz}=4\times 10^{-2}$ N.m.s^2^/rad; $J_{ey}=10$ N.m.s^2^/rad; $k_{xi}=100$ N.m/A; $k_{ye}=80$ N.m/A; $K_{0}=18$ N.m/rad/s; ${\bar{x}}_{r}\left(0\right)=1\times 10^{-4}$ m; ${\bar{y}}_{r}\left(0\right)=2\times 10^{-4}$ m; ${\dot{x}}_{r}\left(0\right)={\dot{y}}_{r}\left(0\right)=0$ m/s; $\alpha \left(0\right)=0.02$ rad; $\beta \left(0\right)=0.015$ rad; $\dot{\alpha }\left(0\right)=\dot{\beta }\left(0\right)=0$ rad/s; $\sigma_{i}\left(0\right)=2.5$ deg; $\sigma_{e}\left(0\right)=3.5$ deg; ${\dot{\sigma }}_{i}\left(0\right)={\dot{\sigma }}_{e}\left(0\right)=0$ rad/s; $\sigma_{ti}=16$ deg; $\sigma_{te}=20$ deg; $\alpha_{c}=\beta_{c}=0$ rad; $\xi_{0}=0.7$; $\omega_{r0}=2.5$ rad/s; ${\dot{\sigma }}_{ic}\left(0\right)={\dot{\sigma }}_{ec}\left(0\right)=0$ rad/s; ${\bar{\alpha }}_{c}\left(0\right)=1.75\times 10^{-4}$ rad; $\beta_{c}\left(0\right)=1.75\times 10^{-4}$ rad.

The coefficients related to the neural networks have the values: $k_{2}=20$; $k_{3}=250$; d=0.1; $b_{W}=1$; $S_{W}=S_{V}=0.1$; $a_{j}={\left[\begin{array}{cccccccccc} 1 & 0.9 & 0.8 & 0.7 & 0.6 & 0.5 & 0.4 & 0.3 & 0.2 & 0.1 \end{array}\right]}^{T}$; $n_{1}=9$ neurons; $n_{2}=10$ neurons; $n_{3}=2$ neurons; $V_{0}=0_{10\times 10}$; $W_{0}=0_{11\times 12}$; $n_{1j}=9$ neurons; $n_{2j}=10$ neurons; $n_{3j}=2$ neurons; $V_{j0}=V_{j}(0)=0_{10\times 10}$; $W_{j0}=W_{j}(0)=0_{11\times 12}$; $\left(j=\bar{2,3}\right)$; $\Gamma_{W}=\Gamma_{V}=0.5$. For the controllers, the following coefficients were chosen: $k_{i1}=30;k_{p1}=31$; $k_{p2}=10$; $k_{d2}=7$; $k_{p3}=6$; $k_{d3}=10$; $k_{i0}=30$; $k_{p0}=25$.

One has performed a simulation using Matlab/Simulink, and the dynamic characteristics were obtained, as depicted in Figure 4 and Figure 5.

The durations of the dynamic regimes are under one second, even under 0.5 s and fall within the imposed limits. The stationary errors are zero.

The dynamic characteristics for the first subsystem (in Figure 2a) for the stabilization and orientation modes are identical (the first six groups of graphs in Figure 4a and the first six groups of graphs in Figure 5a), because the subsystem in Figure 2a is decoupled from those in Figure 2b and Figure 3a for both modes (stabilization and orientation).

The next 12 groups of graphs in Figure 4b and in Figure 5b express the dynamics of the variables of the subsystem in Figure 2b for the stabilization mode and the orientation mode, respectively. The graphs for ${\bar{y}}_{2}$, ${\dot{\bar{y}}}_{2}$, and $y_{2c}$ are effectively identical for the two operating modes, as they are state variables related to the reference model (30), which is not influenced by the subsystem in Figure 3a.

For both modes, the variables in Figure 2b stabilize at zero (${\hat{v}}_{2pd}=$
$=v_{a2}=\varepsilon_{2}=v_{2}={\ddot{y}}_{2}={\ddot{\bar{y}}}_{2}={\dot{y}}_{2}={\hat{\dot{y}}}_{2}={\tilde{y}}_{2}=u_{2}=y_{2}={\left[\begin{array}{cc} 0 & 0 \end{array}\right]}^{T}$). So, the adaptive components compensate for dynamic inversion errors, implicitly $h_{r2}\to {\hat{h}}_{r2}$, and the AMB rotor’s rotation angles, its rates and its angular accelerations cancel out, as required (imposed) in the design stages.

Therefore, the first six groups of graphs, as well as the next twelve groups of graphs, confirm that the linear and angular (precession angle) displacements, along with their rates and accelerations, cancel out, meaning that the AMB rotor’s axis overlaps the inner gimbal axis (the same as the CT axis). The currents applied to the stator coils of the magnetic bearings also become zero ($u_{1}=$
$=u_{2}={\left[\begin{array}{cc} 0 & 0 \end{array}\right]}^{T}$ mA).

The last 15 groups of graphs in Figure 4c and in Figure 5c express the dynamics of the variables belonging to the subsystems in Figure 3a, for the stabilization mode $\left({\bar{y}}_{3c}=y_{3c}={\left[\begin{array}{cc} 0 & 0 \end{array}\right]}^{T}\right)$ and, respectively, for the orientation mode $\left({\bar{y}}_{3c}={\left[\begin{array}{cc} 0 & 0 \end{array}\right]}^{T}\right.$ and $\left.y_{3c}=\sigma_{t}={\left[\begin{array}{cc} 15 & 20 \end{array}\right]}^{T}\right)$. The state variables of the reference model are identical for both regimes, since the reference model is not influenced by either of the two subsystems. Other conclusions are similar to those for Figure 2b; ${\hat{v}}_{3pi}=-{\hat{v}}_{3pd}$, ${\hat{v}}_{a3}=\varepsilon_{3}$, $y_{3}=v_{3}={\left[\begin{array}{cc} 0 & 0 \end{array}\right]}^{T}$, ${\dot{y}}_{3}={\ddot{y}}_{3}={\left[\begin{array}{cc} 0 & 0 \end{array}\right]}^{T}$, $h_{r3}\to {\hat{h}}_{r3}$, $u_{3}={\left[\begin{array}{cc} 0 & 0 \end{array}\right]}^{T}$ A, $\lambda ={\left[\begin{array}{cc} 0 & 0 \end{array}\right]}^{T}$.

## 6. Conclusions

This paper first introduced the DGMSGG’s structure and functions/tasks. Starting from the dynamic models of the DGMSCMG’s subsystems in [8,15], the models of the DGMSGG’s subsystems are established (see (9), (10) and (11)), also taking into account the angular rates of the base (missile), with the predominance of the angular rate of the missile around its longitudinal axis, which generates the angular rate $\omega_{x_{T}}$ around the guidance line. The relative degree in relation to each of the output variables is two.

The three DGMSGG’s subsystems were designed as decoupled systems, using linear dynamic compensators of P.I.D., P.D. and P.I. type. Excepting the subsystems for the automatic control of the dynamics of translation and rotation of the AMB rotor, the subsystem for the control of the gyroscopic gimbal’s dynamics consists of the stabilization contour (with a P.D. type dynamic compensator) and the guidance contour (with a P.I. type dynamic compensator). The controllers for stabilizing the rotations of the AMB rotor and of the gimbals contain, in addition to the linear dynamic compensators of P.D. type, a state observer and a neural network for modeling the adaptive component, this playing the role of compensating the effect of dynamic inversion errors. The design of the adaptive control laws for the three subsystems is based on the works of Calise, for example [30], using the concept of dynamic inversion and the theory of Lyapunov functions, applicable for wide classes of systems described by nonlinear functions, which satisfy the conditions of hypothesis 1 in [29].

The control vector $u_{1}$ contains (see Figure 2a) the currents $i_{x}$ and $i_{y}$, which are applied to the stator coils of the magnetic bearings in the $Ox_{r}$ and $Oy_{r}$ axes of the AMB rotor to cancel its linear displacements, while the vector $u_{2}$ contains the currents $i_{\alpha }$ and $i_{\beta }$, which are applied to the same coils to cancel the angular displacements of the rotor; these currents generate electromagnetic forces and, respectively, correction torques, with the aim of centering (orienting) the axis of the gyroscopic rotor (the axis of the kinetic moment ${\vec{K}}_{0}$) in the direction of the axis of the inner frame (CT‘s axis), in fact, the line of sight. The command vector $u_{3}$ contains the currents $i_{xi}$ and $i_{ye}$, which are applied to the motors for driving the gimbals, for their stabilization and orientation, so that the $Oz_{i}-$ axis (CT’s axis, the line of sight) overlaps with the $Oz_{T}-$ axis (the guide line).

The calculation of the linear dynamic compensators’ parameters is performed by imposing the roots of the characteristic Equations (34), (56), and (71) related to the linear subsystems, resulting from the compensation of the dynamic inversion errors by the adaptive components of the control laws (following the compensation of the nonlinear functions $h_{ri}$ by the functions ${\hat{h}}_{ri}^{-1},i=\bar{2,3}$).

The controlled state variables related to the AMB rotor (components of the $q_{r}-$ vector) are calculated with Formula (6) using the components of the $q_{s}-$ vector (measured by the total linear displacement sensors, arranged on the rotor axes). The rotation angles $\sigma_{i}$ and $\sigma_{e}$ of the gimbals are measured by the angular transducers arranged along the DGMSGG gimbals’ axes (both inner and outer gimbal), as shown in Figure 1a. To determine the state variables ${\dot{x}}_{r}$, ${\dot{y}}_{r}$, $\dot{\alpha }$, $\dot{\beta }$, ${\dot{\sigma }}_{i}$, and ${\dot{\sigma }}_{e}$, the three linear state observers (44), (55), and (54) are used, as well as the state variable vectors ${\dot{\bar{y}}}_{2}={\left[\begin{array}{cc} \bar{\dot{\alpha }} & \bar{\dot{\beta }} \end{array}\right]}^{T}$ and ${\dot{\bar{y}}}_{3}={\left[\begin{array}{cc} {\bar{\dot{\sigma }}}_{i} & {\bar{\dot{\sigma }}}_{e} \end{array}\right]}^{T}$ of the reference models.

According to Formula (73), the missile guidance signals are the very components of the λ− vector (signals provided by the CT for the orientation contour of the system in Figure 3), which is applied to the missile autopilot for its guidance by the parallel approach method (the translation of the guidance line parallel to itself to the point of interception of the target).

The theoretical results are validated by numerical simulation, using Matlab/Simulink models. The dynamic characteristics in Figure 4 and in Figure 5 express superior quality indicators (small overshoots, small stationary errors and also small settling times or small durations of dynamic regimes).

Summarizing the above, we can specify the following elements of novelty and modernity brought by this paper:
A new guidance system structure (guiding head) with a magnetically suspended gyroscope.New nonlinear models describing the interconnected dynamics of the gimbals and rotations of the magnetically suspended gyroscope.Decoupling the three nonlinear dynamics (the dynamics of the gyroscopic rotor’s translations from the dynamics of its rotations and from the dynamics of gimbals’ rotations), the physical coupling terms of the three dynamics being included in the dynamic inversion errors.Deducing the relative degrees of the three output vectors of those three nonlinear dynamics, separating the nonlinear functions ${\hat{h}}_{ri},i=\bar{1,3}$, (which depend only on the input and output vectors of the nonlinear dynamics $h_{ri}$) from the dynamic inversion errors $\varepsilon_{i}$ (which contain nonlinear terms—acting as internal perturbations—and the terms depending on the external perturbations induced by the rotations of the base—the missile).Design of control structures for the three nonlinear dynamics. Using the concept of dynamic inversion, if dynamic inversion errors were absent, the three dynamics would be linear (having transfer matrices $\frac{1}{s^{2}}I_{2}$) and the controllers used would be conventional linear (P.D. or P.I.D. type). However, even in this case (with linear models), coupling errors would lead to stabilization and orientation errors (thus, reducing the accuracy of these subsystems), errors which could not be eliminated, but only diminished.To increase the precision and other quality indicators of the above-mentioned three subsystems’ dynamics, we considered it necessary to compensate for the effects of these dynamic inversion errors (which, in fact, represent disturbances). Their effects are the same as those of some disturbing external inputs, which cannot be eliminated, but can be completely compensated by the adaptive components ${\hat{v}}_{ai}$, which represent the estimation of the dynamic inversion errors $\varepsilon_{i}$. Based on the information collected from the evolution of the vectors ${\hat{v}}_{i}$, from the output vectors $y_{i}$, as well as from the training vectors ${\bar{e}}_{i}$, the neural networks estimate the effects of the internal disturbances of the dynamics of the rotors and gimbals’ rotations. As a result, the equivalent dynamics are reduced to two ideal integrators in series $\left(\frac{1}{s^{2}}I_{2}\right)$ and, implicitly, the deviations ${\tilde{y}}_{i}$ are canceled (zero static errors). So, such structures with adaptive control, from the performance’s point of view, are superior to those based on conventional control.In order to use a minimum number of sensors, both for calculating the linear components of the controllers (the outputs of the linear dynamic compensators) and for calculating the training vectors ${\bar{e}}_{i}$ of the neural networks, linear state observers and reference models were used.A comparison with the performance of other similar system structures can only be made if the same dynamic models are used, having the same numerical values of their physical parameters, but only on the condition of using suitable estimators for disturbances.

## Figures and Tables

**Figure 1 micromachines-16-00245-f001:**
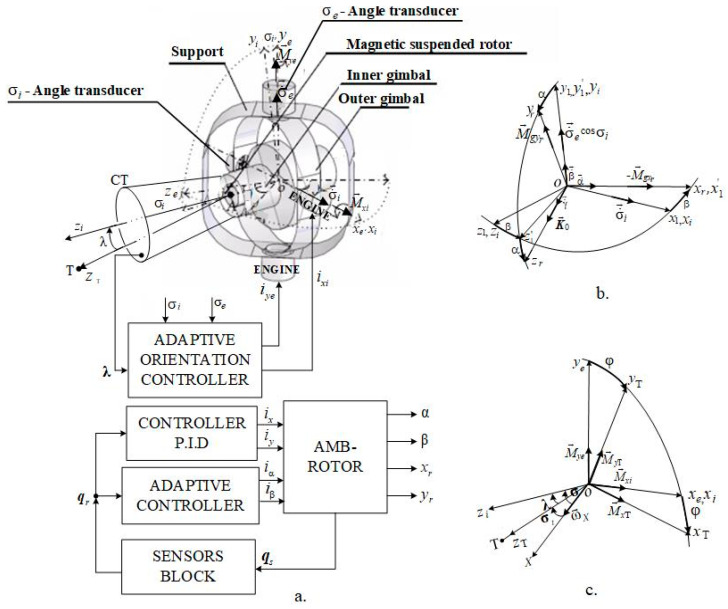
Frames related to the dynamic components of DGMSGG, rotation angles, angular rates, and correction moments. (**a**) DGMSGG’s architecture; (**b**) gyroscopic rotor’s centering; (**c**) overlapping the line of sight over the guidance line.

**Figure 2 micromachines-16-00245-f002:**
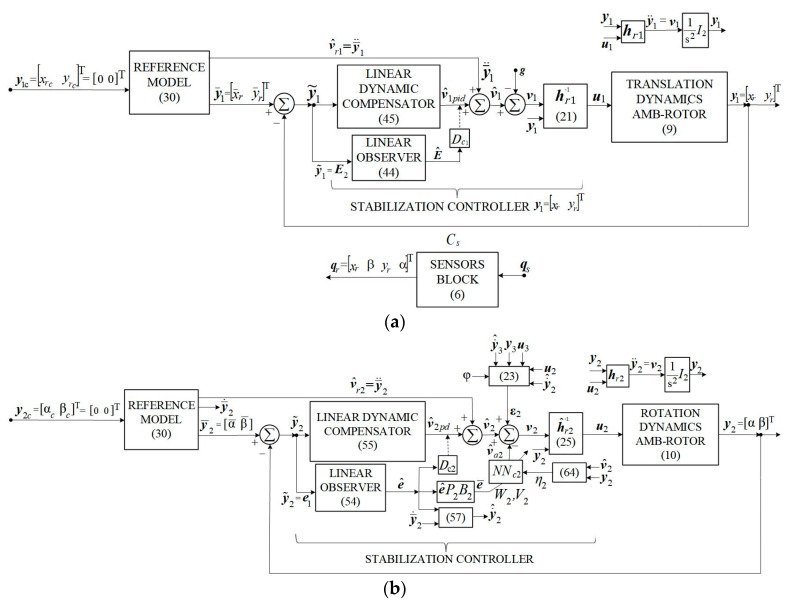
Subsystems for automatic control of gyroscopic rotor’s dynamics (**a**) translation dynamics control; (**b**) rotation dynamics control.

**Figure 3 micromachines-16-00245-f003:**
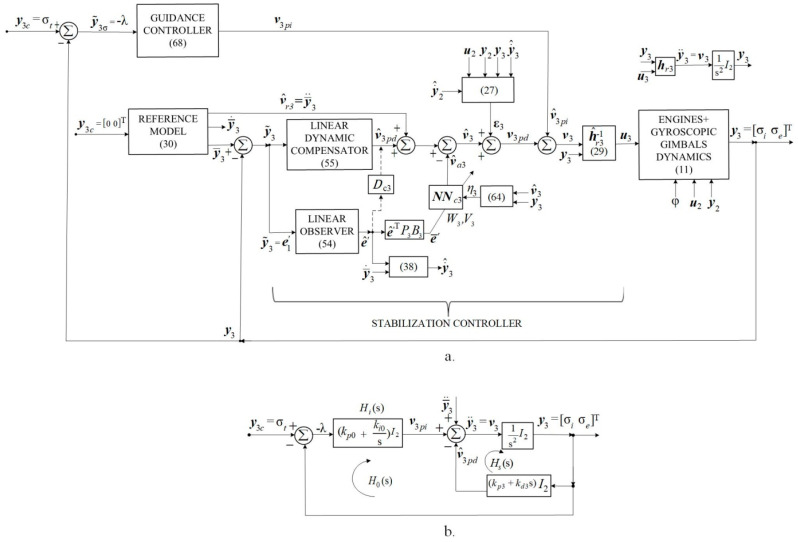
Automatic control system of gyroscopic gimbals’ nonlinear dynamics; (**a**) the complete block diagram; (**b**) linear subsystem’s block diagram.

**Figure 4 micromachines-16-00245-f004:**
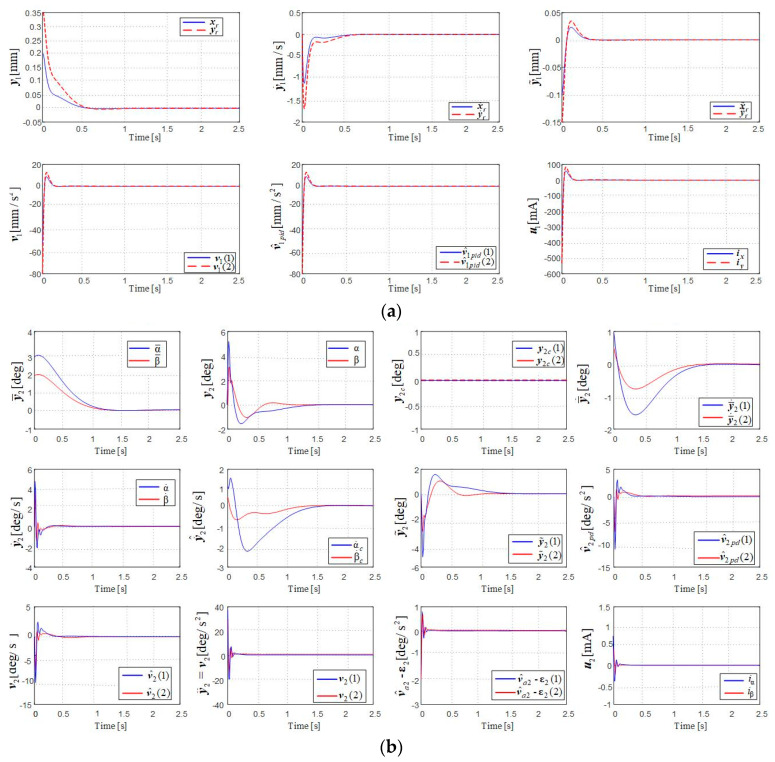
Dynamic characteristics of the DGMSGG in stabilization mode: (**a**) of the subsystem in Figure 2a; (**b**) of the subsystem in Figure 2b; (**c**) of the subsystem in Figure 3a.

**Figure 5 micromachines-16-00245-f005:**
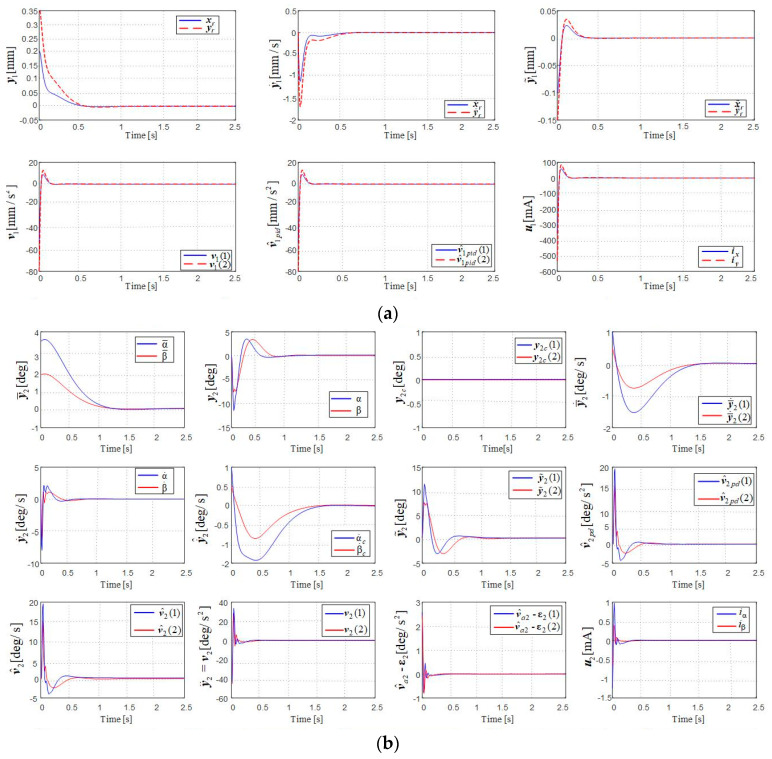
Dynamic characteristics of the DGMSGG in guidance (orientation) mode: (**a**) of the subsystem in Figure 2a; (**b**) of the subsystem in Figure 2b; (**c**) of the subsystem in Figure 3a.

## Data Availability

The original contributions presented in this study are included in the article. Further inquiries can be directed to the corresponding authors.

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
