# Peer review of "Guidance Gyro System with Two Gimbals and Magnetic Suspension Gyros Using Adaptive-Type Control Laws"

_micromachines, 2025, doi:10.3390/mi16030245_

Round 1
Reviewer 1 Report
Comments and Suggestions for Authors
This paper proposes an engineering-practical solution for the design and control of dual-frame magnetically suspended gyroscope systems. The integrated approach combining dynamic inversion with neural networks offers novel insights for decoupling control in complex dynamic systems. However, insufficient depth in literature review, low transparency in methodological details, and limitations in experimental validation undermine the academic rigor of this work. Supplementing comparative analyses, refining technical descriptions, and standardizing writing conventions would significantly enhance both its academic impact and engineering reference value.
1. Redundant use of "with".
2. Lack of explicit distinction between DGMSCMG (actuator) and DGMSCG (guidance system) in abstract
3. Incomplete coverage of recent research findings in the literature review section
4. Please provide specific metrics for”The theoretical results are validated by numerical simulation, using 481 Matlab/Simulink models. The dynamic characteristics in Figures 4 and 5 express superior 482 quality indicators (small overshoots, small stationary errors and also small settling times 483 or small durations of dynamic regimes).”
5. The specific implementation steps of Dynamic Inversion are not clearly demonstrated, such as how to select a reference model or handle nonlinear coupling terms.
6. The architecture of the neural network (e.g., number of layers, neurons, activation functions) and its training process (e.g., loss function, dataset sources) lack detailed descriptions.
7. The simulation results only demonstrate the performance of the proposed method (e.g., overshoot, steady-state error), but lack comparisons with baseline methods such as traditional PID control and sliding mode control. This omission prevents the validation of its superiority.
8. The parameter sensitivity analysis is missing. For instance, the impact of controller gains or neural network weight initialization on system stability remains undiscussed.
9. The distinction between "physical decoupling" and "dynamic decoupling" has not been thoroughly discussed. This omission obscures the technical contributions.
10. Some lengthy sentences exhibit complex logical structures. This results in increased comprehension difficulty.
11. Inconsistent terminology usage is observed.
Comments on the Quality of English Language
The English Language is good.
Author Response
Please read the attached document.

Reviewer 2 Report
Comments and Suggestions for Authors
The authors attempt to optimize DGMSGG. In terms of research motivation, from the introduction, decoupling of the rotor model and corresponding errors are mentioned. However, the proposed system is not confirmed by any results. The manuscript lacks experimental data and only includes simulation results, which are presented in Fig. 4 and Fig. 5. To my surprise, these two results are not only devoid of any explanation or analysis (just a one-sentence introduction), but they are also incomplete (e.g., Fig. 4 overlaps with Fig. 3 in layout and the display is incomplete). This makes the scientific soundness, the presentation of the results, and the reliability of the proposed method very poor.
In addition, the manuscript focuses on deriving mathematical models of control systems. However, the diagram of the control system is fuzzy and incomplete (e.g., Fig. 4 overlaps with Fig. 3), which affects the readability. The derivation is verbose and fails to highlight the innovation of this manuscript. Some equations have typesetting issues (e.g., the dot "." after φ in Eq. (8) ). I think the authors should simplify the derivation process, present the core ideas of this manuscript through fewer equations, and relocate the redundant derivation to the appendix. This will better present the ideas and innovations of this work. Furthermore, the diagrams can also be simplified. For example, in Fig.1b, the same physical quantity and its different forms can be considered to plot only one of them to reduce the excessive variables.
Author Response
Please read the attached document.

Reviewer 3 Report
Comments and Suggestions for Authors
I have no comments. A very well elaborated article, complex due to the scientific theme addressed, with a very good logic supported by an advanced mathematical apparatus. I believe that the work has a high scientific level and the use of gyroscopes in aeronautical engineering. Well-structured, well-proportioned article, we also note the extensive and high-quality bibliography. Congratulations to the authors.
Author Response
Please read the attached document.

Reviewer 4 Report
Comments and Suggestions for Authors
Recommendations: Explain technical terms at their first mention (e.g., "dynamic inversion" or "parallel approach guidance"). Potentially confusing terms in the text:
- Dynamic inversion – The concept may not be clear to all readers without further elaboration on how it applies to the control systems discussed.
- Parallel approach guidance – Needs an explanation of how this method works in missile guidance systems.
- Double Gimbal Magnetic Suspension Gyro-System for Guidance (DGMSGG) – The acronym is explained, but a brief description of its significance or unique features could be added.
- Active Magnetic Bearings (AMB) – While the acronym is expanded, the underlying mechanism and its advantages could be clarified.
- Sight line and guidance line – These terms are used frequently, but their specific meaning and distinction may not be immediately evident to all readers.
- Precession motion – Though a technical term, it could benefit from a simple definition or context within the gyro-system dynamics.
- Decoupled dynamics – The term could be expanded to explain its importance in the context of the system's control architecture.
- State observers – Readers might need clarification on their role in adaptive control systems.
Author Response
Please, read the attached document.

Round 2
Reviewer 1 Report
Comments and Suggestions for Authors
The authors have revised the manuscript in a satisfactory way and the work can be published.
Author Response
Dear Reviewer,
Thank you for agreeing to review our manuscript; also, thank you for the remarks and
suggestions made to improve the quality of our paper.
Kindest regards,
R. L, C.- A. M. and A.-N. T.
Submission date: 12 February 2025
Reviewer 2 Report
Comments and Suggestions for Authors
As for the simulation results in Fig. 4 and Fig. 5, thanks to the authors for the supplementary explanations at the end of the Conclusion. In my opinion, this part would be better placed after Line 451-452 to improve readability. Additionally, I think there is a typo. Should "the last 12 groups" in Line 528 be changed to "the last 15 groups"?
The authors should comprehensively revise their manuscript against the instructions of the journal (https://www.mdpi.com/journal/micromachines/instructions#manuscript). The readability of the manuscript in its current form is still poor:
- Although the authors added a discussion of the simulation results, it still lacks a concise and precise description of the results.
- Fig. 4 and Fig. 5 contain excessive subgraphs and are full of a large number of variables and symbols, which greatly reduces readability. The authors could consider splitting Fig. 4 and Fig. 5 according to the performance classification (e.g., the authors' supplementary description could be categorized into dynamic characteristics for different subsystems) to facilitate comparison for readers.
The authors state that the results are not available, which the reviewer understands. However, the existing simulation results in the manuscript do not demonstrate that the proposed system outperforms conventional methods. Due to the insufficient comparison with conventional methods and the lack of a comparative analysis with similar systems, it is difficult to confirm the novelty and advantages of this work.
Author Response
Please, read the attached file.
